# Accuracy of FIB-4 to Detect Elevated Liver Stiffness Measurements in Patients with Non-Alcoholic Fatty Liver Disease: A Cross-Sectional Study in Referral Centers

**DOI:** 10.3390/ijms232012489

**Published:** 2022-10-18

**Authors:** Mauro Viganò, Nicola Pugliese, Federica Cerini, Federica Turati, Vincenzo Cimino, Sofia Ridolfo, Simone Rocchetto, Francesca Foglio, Maria Terrin, Carlo La Vecchia, Maria Grazia Rumi, Alessio Aghemo

**Affiliations:** 1Hepatology Division San Giuseppe Hospital, MultiMedica IRCCS, Università degli Studi di Milano, Via San Vittore 12, 20123 Milan, Italy; 2Division of Internal Medicine and Hepatology, Department of Gastroenterology, IRCCS Humanitas Research Hospital, 20089 Rozzano, Italy; 3Department of Clinical Sciences and Community Health, Università degli Studi di Milano, Via San Vittore 12, 20123 Milan, Italy; 4Department of Biochemical and Clinical Science L. Sacco, Università degli Studi di Milano, Via San Vittore 12, 20123 Milan, Italy; 5Department of Biomedical Sciences, Humanitas University, 20072 Pieve Emanuele, Italy

**Keywords:** NAFLD, FIB-4, liver stiffness, noninvasive test, diabetes mellitus, obesity, metabolic syndrome

## Abstract

The identification of advanced fibrosis by applying noninvasive tests is still a key component of the diagnostic algorithm of NAFLD. The aim of this study is to assess the concordance between the FIB-4 and liver stiffness measurement (LSM) in patients referred to two liver centers for the ultrasound-based diagnosis of NAFLD. Fibrosis 4 Index for Liver Fibrosis (FIB-4) and LSM were assessed in 1338 patients. A total of 428 (32%) had an LSM ≥ 8 kPa, whereas 699 (52%) and 113 (9%) patients had an FIB-4 < 1.3 and >3.25, respectively. Among 699 patients with an FIB-4 < 1.3, 118 (17%) had an LSM ≥ 8 kPa (false-negative FIB-4). This proportion was higher in patients ≥60 years, with diabetes mellitus (DM), arterial hypertension or a body mass index (BMI) ≥ 27 kg/m^2^. In multiple adjusted models, age ≥ 60 years (odds ratio (OR) = 1.96, 95% confidence interval (CI) 1.19–3.23)), DM (OR = 2.59, 95% CI 1.63–4.13), body mass index (BMI) ≥ 27 kg/m^2^ (OR = 2.17, 95% CI 1.33–3.56) and gamma-glutamyltransferase ≥ 25 UI/L (OR = 2.68, 95% CI 1.49–4.84) were associated with false-negative FIB-4. The proportion of false-negative FIB-4 was 6% in patients with none or one of these risk factors and increased to 16, 31 and 46% among those with two, three and four concomitant risk factors, respectively. FIB-4 is suboptimal to identify patients to refer to liver centers, because about one-fifth may be false negative at FIB-4, having instead an LSM ≥ 8 KPa.

## 1. Introduction

Nonalcoholic fatty liver disease (NAFLD) is the most prevalent liver disease worldwide, with increasing figures across the globe as a consequence of the obesity pandemic, population ageing and changing dietary habits [1]. Overall, between 24 and 28% of the general population suffers from this condition, with a prevalence that can reach 60–70% in patients with obesity/diabetes [2,3,4,5,6]. Although growing evidence suggests that liver-related complications can be frequently observed in patients without advanced fibrosis/cirrhosis [7], the identification of patients with liver fibrosis is still a key component of the diagnostic algorithm for NAFLD [8,9]. Indeed, the risk of complications and death is directly linked to the fibrosis stage, suggesting the rapid identification and referral to liver centers for this group of patients [9,10,11,12,13]. Developing and applying noninvasive tests (NITs) for fibrosis assessment and risk stratification in patients with NAFLD has been a focus of research in the last years. NITs are cost-effective, easy to use and thus can be applied at the population level to identify which patients with NAFLD need a referral [8,9,10,11,12,13,14].

When looking at primary care, most guidelines suggest using FIB-4 as the first discriminating NIT [9,15]. FIB-4 is a simple test developed for viral infections, which includes age, platelet count and alanine aminotransferase (ALT) [16]; a cutoff of <1.3 has been shown to confidently rule out the presence of significant fibrosis [17,18]. In patients with elevated FIB-4, a second test, such as Enhanced Liver Fibrosis (ELF) or liver stiffness measurement (LSM) by Fibroscan, improves the accuracy and prediction of significant fibrosis [19]. The LSM has a good accuracy to rule in or rule out fibrosis but cannot be advocated in a primary care/non-specialist setting, mainly for the limited availability caused by a dedicated device [20,21,22,23]. Whether recommendations focusing on the patient’s stratification and referral are being followed at the national level is largely unknown, and the accuracy of FIB-4 in this population is still unknown as a large international study reported non-negligible rates of false-negative FIB-4 showing fibrosis by LSM [24].

We designed a multicenter cross-sectional study including all consecutive patients of two referral liver centers in Milano, Italy, for an ultrasound-based diagnosis of NAFLD. Our aims were (I) to assess if referred patients follow the international guidelines, (II) assess the concordance between the FIB-4 and LSM and (III) understand the factors associated with false-negative FIB-4 results which might delay referral to liver centers.

## 2. Results

During the study period, a total of 1338 patients were enrolled for this study. Their characteristics are summarized in Table 1.

The mean age was 59 years (range 18–88), most (769, 57%) patients were males and the most frequent ethnicity was Caucasian (1197, 89%); 5% of the patients were Hispanic. The median BMI was 28 kg/m^2^; 1065 (80%) patients were overweight. A total of 441 patients (32%) had DM, 657 (49%) had arterial hypertension and 602 (45%) were dyslypidemic. The median ALT levels were below the upper limit of normal (ULN), but 514 (38%) had ALT levels > ULN. The median LSM (assessed with an M and XL probe in 563 (42%) and 775 (58%) patients, respectively) was 6.0 (interquartile range: 4.7–9.0, range 2.1–75.0).

### FIB-4 and Transient Elastography

Overall, 699 patients (52%) had an FIB-4 < 1.3, 113 (9%) had an FIB-4 > 3.25 suggestive of advanced chronic liver disease, while 526 (39%) had an intermediate FIB-4 (1.3–3.25). In terms of the LSM by elastography, 910 patients (68%) had an LSM value < 8 kPa, 235 (18%) had an LSM value of 8–11.9 kPa, while 193 (14%) had an LSM ≥12 kPa.

While 581 (83%) patients with an FIB-4 < 1.3 showed LSM values below the 8 KPa threshold, 118 (17%) patients had an LSM suggestive of intermediate–advanced fibrosis. More precisely, 3% of patients with an FIB-4 < 1.3 showed an LSM indicative of severe fibrosis (>12 kPa), while 14% of them showed an LSM compatible with intermediate fibrosis (8–12 kPa). Among patients with an intermediate FIB-4 (1.3–3.25), 60% had an LSM < 8 kPa, 22% had an intermediate LSM and 18% had an LSM >12 kPa. Among patients with an FIB-4 > 3.25, 11% showed an LSM indicative of a normal liver (<8 kPa), 18% showed an LSM indicative of mild–moderate fibrosis (8–12 kPa) and 71% showed an LSM indicative of advanced fibrosis (>12 kPa). 

The distribution of selected characteristics of NALFD patients with an FIB-4 < 1.3 is provided in Table 2.

The mean age was 52 years and 57% of the patients were males. DM, arterial hypertension and dyslipidemia were reported by 25, 37 and 40% of patients. The median BMI was 28 kg/m^2^ and 25% of the patients had a BMI ≥ 31 kg/m^2^.

Among 699 NAFLD patients with an FIB-4 score < 1.3, 118 (17%) had an LSM ≥8 kPa. The proportion was higher in the elderly (26% in patients ≥ 60 years), those with DM (31%), hypertension (24%) and patients with a BMI ≥ 27 kg/m^2^ (22%) (Table 3). In addition, an LSM ≥ 8 kPa was detected more frequently in patients with gamma-glutamyltransferase (GGT) ≥ 25 UI/L (19%) than in those with a GGT < 25 UI/L (8.9%). In unadjusted analyses, age, DM, hypertension, BMI and elevated GGT were directly associated with increased LSM; no association was found with dyslipidemia, bilirubin, ALP, total and high-density lipoprotein cholesterol and triglycerides. Similar results were found in the multiple adjusted analysis, but the association with hypertension was no longer significant. The risk was increased over 2.5 times in patients with a GGT ≥ 25 UI/L (odds ratio, OR = 2.68, 95% confidence interval, CI 1.49–4.84) and with DM (OR = 2.59, 95% CI 1.63–4.13), and the OR was 2.17 (95% CI 1.33–3.56) in patients with a BMI ≥27 kg/m^2^ and 1.96 (95% CI 1.19–3.23) in those older than 60 years of age. Male patients had a 33% non-significantly increased risk of an increased LSM compared to females.

The proportion of patients with an increased LSM was around 6% in the subgroup of patients with none or one of the risk factors previously identified (age ≥ 60 years, BMI ≥ 27 kg/m^2^, DM and GGT ≥ 25 UI/L) and increased to 16, 31 and 46% among patients with two, three and four risk factors, respectively (Table 4). An over 14-fold excess risk was found for the presence of all the four identified risk factors compared to 0–1.

The results for selected combinations of the risk factors are presented in Appendix A
Table A1. Among the patients with age ≥ 60, DM and GGT ≥ 25 UI/L, 23 (44.2%) had an LSM ≥ 8 kPa, and the corresponding OR was 4.64 (95% CI 2.53–8.53). The corresponding figures in the patients with age ≥ 60, BMI ≥ 27 kg/m^2^ and GGT ≥ 25 UI/L were 29 (39.7%) and 3.26 (95% CI 1.88–5.64) and in those with age ≥ 60, DM and BMI ≥ 27 kg/m^2^ were 23 (39.7%) and 3.96 (95% CI 2.16–7.23).

The predictive model formulation is given in the Appendix B. Each unit increase in age, BMI and GGT increases the natural logarithm of the odds of an LSM ≥ 8 kPa, respectively, by 0.022, 0.127 and 0.007; this was increased by 1.03 in diabetics. The model AIC was 548.2, and the area under the receiver operating characteristic curve (AUC-ROC was) 0.743, indicating a reasonable accuracy of the model in discriminating between patients with and without an increased LSM (Figure 1A). The sensibility and specificity for different predicted probability thresholds are provided in Figure 1A. With a probability threshold of 0.166, equal to the observed proportion of patients with an LSM ≥ 8 kPa in our study sample (i.e., 113/680), the sensitivity and specificity were, respectively, 0.68 and 0.70. The bootstrap internal validation revealed minimal predictive optimism, with an optimism-adjusted AUC-ROC for the model of 0.733 (95% CI 0.692–0.786), close to the unadjusted accuracy measure. A visual inspection of the calibration plot showed an acceptable agreement between the predicted and observed probabilities (Figure 1B).

## 3. Discussion

Patients with NAFLD may progress to NASH and liver fibrosis, and those with advanced fibrosis or cirrhosis trend toward more complications of liver failure and hepatocellular carcinoma compared to those without. The identification of such patients by applying NITs for fibrosis assessment and risk stratification is still a key component of the diagnostic algorithm of NAFLD, because the stage of liver disease is the strongest predictor for long-term clinical outcomes [10,11]. The FIB-4, NAFLD Fibrosis Score (NFS) and AST/platelet ratio index (APRI) are models commonly used for detecting fibrosis [14], having demonstrated the ability to predict liver-related morbidity and mortality, with a level of performance that met or exceeded that of a liver biopsy [25].

FIB-4 has been developed in high-prevalence populations at secondary centers [16,17], but its utility as screening in comparison to the LSM by Fibroscan has not been properly evaluated so far. The FIB-4 score was initially tested with patients with hepatitis C virus (HCV) and Human Immunodeficiency Virus (HIV) infection.

In our study, we concomitantly assessed the FIB-4 and LSM in all consecutive patients that were referred at our liver centers for the first time for an ultrasound-based diagnosis of NAFLD. We reported an LSM ≥8 kPa in one-third of cases, of which nearly 50% were with an LSM > 12 kPa, yet with the presence of a significant proportion of false-negative FIB-4 results (<1.3) in subjects with an elevated LSM (≥8 kPa). Most importantly, this proportion was higher in subjects at increased risk of liver disease, i.e., ≥60 years, with DM, arterial hypertension or a BMI ≥ 27 kg/m^2^ with a percentage of the false-negative FIB-4 that increased with increasing risk factors, reaching 46% among those with four concomitant risk factors.

This is one of the largest cohorts of NAFLD patients at risk of liver disease, evaluating the concordance between the FIB-4 and LSM where all co-factors of liver damage were excluded, especially alcohol consumption.

Although the reference standard for an accurate assessment of NASH or the fibrosis stage in NAFLD is a liver biopsy, this invasive procedure is resource intensive, it carries a risk of severe complications, it presents the risk of high sampling variability and of inaccurate staging and, above all, cannot be performed on a large number of patients [26,27,28,29]. On the other hand, there is growing promise in risk stratification using NITs for identifying NAFLD patients more likely to develop severe liver events. Using more reliable markers than a liver biopsy would help circumvent the limitations of this procedure in stratifying patients. Moreover, accurate prognostic markers can eventually replace a biopsy and aid clinical decision making, as well as facilitate the recruitment of patients more likely to benefit from participation in clinical trials.

According to the recent EASL CPG on NITs, in patients with metabolic co-factors at risk of liver disease, the FIB-4 is useful for identifying patients requiring a referral to the specialist liver clinics. More precisely, patients with an FIB-4 < 1.3 do not need a referral and should be re-tested in 1–3 years, whereas those with an FIB-4 ≥ 1.3 should be considered for a Fibroscan and subsequently re-stratified accordingly [9].

Despite an established diagnostic performance, there is a limited understanding of the relative merits of the prognostic ability of the FIB-4 and its comparability to a Fibroscan. While many studies have assessed the diagnostic performance of these markers in reference to a liver biopsy, and some have validated the FIB-4 as a predictor, only recently a multicenter study described the accuracy of the FIB-4 to detect an elevated LSM in 5129 patients (3979 from general population cohorts and 1150 from at-risk cohorts) [24]. Considering only those from the risk cohort due to alcohol, DM or obesity, 29% had an LSM ≥ 8 kPa (including 14% with an LSM > 12 kPa) with 8% of false-positive FIB-4 results (11% in DM). Although it is not possible to analyze the NAFLD patients only, the paper by Graupera et al. [24] has data similar to ours, underlining the risk of not referring patients to liver centers, because a large group of patients with an elevated LSM, and therefore with probable advanced liver disease, may remain unidentified by relying solely on the FIB-4.

With the limitation of having compared noninvasive indirect fibrosis testing (FIB-4) with “direct” fibrosis testing (LSM), our data are similar to that reported by Foschi et al. on the lack of association of the LSM with both the AST/ALT and FIB-4 scores [30].

If further confirmed in prospective studies, the recommended strategy of performing an FIB-4 in a risk population to select those patients to refer should be changed to reduce the significant rate of patients with advanced liver disease which may not be referred otherwise.

An ideal study to answer this question should provide the simultaneous execution of an FIB-4, LSM and liver biopsy, as a reference standard, in a large group of patients with a uniform distribution of the different fibrosis stages. However, such a study is difficult to perform because a liver biopsy is reserved in most cases to subjects with evidence of more advanced liver disease or to identify patients for clinical trials for NAFLD/NASH treatments. While waiting for these data, we can rely on Fibroscan as a surrogate for liver fibrosis by modifying the FIB-4 thresholds in order to maximize its sensitivity in identifying an LSM ≥ 8 kPa, thus having a high negative predictive value (NPV). This threshold change should be considered at least in patients with two or more of the identified risk factors of FIB-4 failure, i.e., elderly, with DM or those who are overweight (or, more simply, in those with metabolic syndrome), to avoid that patients with advanced disease are not referred to referral centers. It is in fact well known that FIB-4 is influenced by age and BMI and performs poorly within obese patients and patients under 35 or over 65 years of age [31,32].

To overcome these obstacles, a new noninvasive fibrosis score (the Hepamet fibrosis score) was developed and validated among 2452 NAFLD patients. The Hepamet fibrosis score, which incorporates demographic, anthropometric and simple laboratory parameters to predict advanced fibrosis and cirrhosis, seems to provide improved accuracy and fewer undetermined results compared to the FIB-4, together with a test performance that is not affected by age or BMI [33]. The sequential use of a cheap, blood-based test first, followed by a confirmatory LSM, increases the number of detected patients with cirrhosis, reduces the proportion of futile referrals and is highly cost-effective [34,35,36].

Although the current CPG on NITs recommend a re-check of the FIB-4 at 1–3 years in those below the 1.3 cutoff, it is likely that not all patients will be re-evaluated within the suggested period and that some may develop complications, or not be considered for drugs for fatty liver disease in the future [9,37]. For this reason, and in accordance with the conclusions of the study by Graupera et al., we recommend that patients at greater risk of liver fibrosis need to be considered for an LSM independently from the FIB-4 values [24].

On the other hand, we also observed a significant proportion of false-positive FIB-4, having an FIB-4 > 1.3 but an LSM < 8 kPa: 51% in our cohort and 59% in the Graupera study [24]. Although we believe that this is less worrying compared to false-negative results, these false-positive patients should not have been sent to a referral.

This study has some limitations because the number of included patients is limited considering the NAFLD prevalence data. Moreover, the number of included patients is different between the two referral centers: although they are both tertiary centers, one is in an urban area, while the other is in a suburban area. The retrospective nature of the study might adversely affect the reliability of the results: a prospective and multicentric study should be planned in order to overcome this bias. Another limitation is the absence of data about liver fibrosis assessed by liver biopsy for the included patients: although it is the gold standard, its invasiveness does not allow its use on a large scale.

Finally, as for the prediction model, we did not perform an external validation, which would test the generalizability of the results outside of our source population. However, for an internal validation, we used the bootstrapping method, which was found to be the best method for producing efficient estimates that have a low bias and low variability [38].

## 4. Materials and Methods

We included patients referred for a diagnosis of NAFLD based on abdominal ultrasound examination from January 2016 to December 2021 at IRCCS Humanitas Research Hospital (*n* = 221) and San Giuseppe IRCCS MultiMedica Hospital in Milan (*n* = 1117).

All patients underwent routine blood test and transient elastography within 30 days from the first visit. Patients with incomplete data were excluded from analysis. Patients were also excluded if they had evidence of secondary causes of chronic liver disease, including hepatitis B or C, or autoimmune hepatitis. Moreover, we excluded patients with significant alcohol consumption (≥30 g/day for men and ≥20 g/day for women) and those receiving hepatotoxic medications. In addition, patients who could not undergo Fibroscan examination and patients with evidence of advanced or decompensated liver disease were also not included (Figure 2).

Demographic, clinical and biochemical data were collected at the first visit. The demographic data of interest were age, sex and ethnicity. The clinical characteristics included BMI and presence of comorbidities, such as DM, arterial hypertension and dyslipidemia. Biochemical data of interest were measured by conventional laboratory test and included platelet count, liver damage and function tests (aspartate aminotransferase (AST), ALT, alkaline phosphatase, GGT, total bilirubin, international normalized ratio (INR), albumin), lipid profile (total cholesterol, high-density lipoprotein, triglycerides), serum fasting glucose.

For each patient, FIB-4 index was determined as follows (age in years, ALT and AST in IU/L and platelet count in 10^9^/L):(1)$$\frac{age\;\left(y\right)x\;AST\;\left(U/L\right)}{platelets\;count\;x\;\sqrt{ALT\left(U/L\right)}}$$

Cutoffs of <1.30 and of >3.25 were selected for their accuracy to rule out and rule in advanced fibrosis.

Moreover, each patient underwent vibration-controlled transient elastography by Fibroscan Mini+ 430, manufactured by Echosens (Paris, France). Transient elastography was performed by two expert physicians (MV and AA) following the manufacturer’s guidelines. A success rate higher than 70% was achieved for all patients included in the study. Different probes were used according to patient’s BMI: patients with a BMI < 30 kg/m^2^ were evaluated with M probe, while patients with a BMI > 30 kg/m^2^ were evaluated with XL probe. The chosen cutoffs were LSM ≥8 kPa to identify patients with presumed liver fibrosis requiring further evaluation and >12 kPa as the optimal cutoff for ruling in advanced liver fibrosis (≥stage 3) [20,39].

### Data Analysis

Continuous variables were checked for normal distribution by visual inspection of histograms and Kolmogorov–Smirnov tests. Normal and non-normal distributed continuous variables were presented, respectively, as mean ± standard deviations and median plus interquartile range; categorical data were presented as absolute frequencies and percentages. The association of individual demographic, laboratory, metabolic and clinical factors with the binary outcome of increased LSM, i.e., LSM ≥ 8 kPa, in NAFLD patients with FIB-4 < 1.3 was assessed using logistic regression models, with estimation of the OR with 95% CI. First, we ran unadjusted models, one for each of the factors considered. Then, we fitted a unique multivariable logistic regression model, including simultaneously as independent variables those significantly associated with the outcome (i.e., LSM > 8) in the unadjusted analyses. Patient sex was also included in the multivariable model.

A logistic regression model, adjusted for sex, was also used to assess the association with a variable representing the number of risk factors for increased LSM (i.e., age ≥ 60 years, BMI ≥ 27 kg/m^2^, DM and GGT ≥ 25 UI/L; variable range: 0–4), as identified in the analysis described above. We also estimated the ORs of increased LSM for selected combinations of risk factors.

In a secondary complementary analysis, we developed a model for prediction of increased LSM in NAFLD patients with FIB-4 score < 1.3. We did not partition our dataset into derivation and testing subsets, but we used the full study population in the model development process and conducted internal model validation by bootstrapping techniques to quantify the amount of over-optimism of the model and to adjust the model’s predictive accuracy for over-optimism [40]. Overall, 200 bootstrap samples, of the same size as the original dataset, were generated by drawing with replacement. Age, sex, DM, BMI and GGT were selected as predictors, based on results of unadjusted and multiple-adjusted analyses and clinical significance. In a sensitivity analysis, we obtained the same selection of variables when considering more candidate predictors (i.e., age, sex, DM, BMI, GGT, hypertension and dyslipidemia) and applying a stepwise selection procedure with variable entry and retention criteria of *p* < 0.20. We assessed model performance in terms of discrimination with the UC-ROC and model calibration by comparing the average model prediction with the observed proportion of increased LSM across deciles of risk (calibration plot). For internal model validation, we applied bootstrapping with 200 bootstrap samples (of the same size as the original population and formed by drawing randomly with replacement from the study database) and calculated the optimism-adjusted AUC-ROC according to the Harrell’s method [41]. The 95% CI of the optimism-corrected AUC-ROC was calculated according to the location-shifted bootstrap confidence intervals method [42].

*p*-values were two-tailed, with *p* < 0.05 considered statistically significant.

All the analyses were performed using the SAS software, version 9.4 (SAS Institute, Inc., Cary, NC, USA).

## 5. Conclusions

In patients with NAFLD, the early identification of those with advanced fibrosis by applying NITs for a fibrosis assessment and risk stratification is still a key component of the diagnostic algorithm of NAFLD. Our data show a significant proportion of false-negative FIB-4 results that could result in the lack of a referral to liver centers. Therefore, we should be cautious in advocating FIB-4 as the only score for specialist referral, especially in NAFLD patients with metabolic risk factors that are intrinsically associated with a worse prognosis.

## Figures and Tables

**Figure 1 ijms-23-12489-f001:**
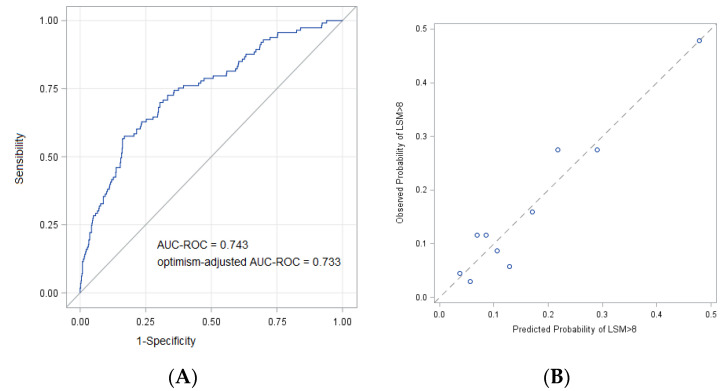
Performance of the prediction model for LSM ≥8 kPa in patients with NAFLD and FIB-4 < 1.3. (**A**): Receiver operating characteristic curve; (**B**): calibration plot displaying observed versus model-predicted probabilities across deciles of risk.

**Figure 2 ijms-23-12489-f002:**
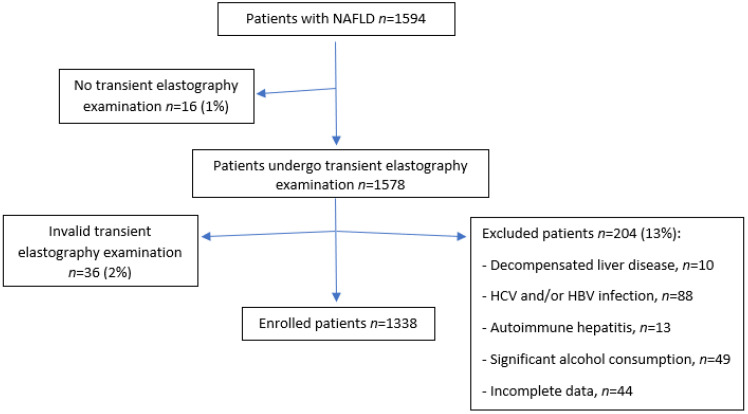
Flow chart with numbers of excluded patients.

**Table 1 ijms-23-12489-t001:** Demographic and clinical features of the overall cohort (*n* = 1338).

	*n* = 1338
Age, mean (std)	59 (13)
Male sex, *n* (%)	769 (57)
Ethnicity, *n* (%)	
Caucasian	1197 (89)
Hispanic	67 (5)
Asian	37 (3)
African	37 (3)
BMI (kg/m^2^), median (IQR)	28 (25.7–31.2)
Overweight, *n* (%)	1065 (80)
DM, *n* (%)	441 (32)
Hypertension, *n* (%)	657 (49)
Dyslipidemia, *n* (%)	602 (45)
Platelets (10^3^/μL), median (IQR)	227 (181–276)
Total bilirubin (mg/dL), median (IQR)	0.7 (0.50–0.97)
AST (UI/L), mean (IQR)	29 (22–42)
ALT (UI/L), median (IQR)	36 (23–61)
ALP (UI/L), median (IQR)	80 (70–129)
GGT (UI/L), median (IQR)	49 (26–102)
INR, median (IQR)	1 (1–1.1)
Albumin (g/dL), median (IQR)	4.2 (4.0–4.4)
Total cholesterol (mg/dL), median (IQR)	190 (162–220)
HDL cholesterol (mg/dL), median (IQR)	49 (41–60)
Triglycerides (mg/dL), median (IQR)	126 (92–175)
Glucose (mg/dL), median (IQR)	102 (92–121)

ALT: alanine aminotransferase; AST: aspartate aminotransferase; BMI: body mass index; DM: diabetes mellitus; GGT: serum gamma-glutamyltransferase; HDL: high-density lipoprotein; INR: international normalized ratio; IQR: interquartile range; std: standard deviation.

**Table 2 ijms-23-12489-t002:** Distribution of selected characteristics * of 699 NAFLD patients with FIB-4 < 1.3.

	*n* = 699
Age, mean (SD)	52 (12.2)
Male, *n* (%)	398 (56.9)
DM, *n* (%)	172 (25)
Hypertension, *n* (%)	259 (37)
Dyslipidemia, *n* (%)	281 (40)
BMI (kg/m^2^), median (IQR)	28 (25–31)
GGT (UI/L), median (IQR)	40 (24–87)
Bilirubin (mg/dL), median (IQR)	0.7 (0.5–0.9)
ALP (UI/L), median (IQR)	85 (68–114)
Albumin (g/dL), median (IQR)	4.3 (4.0–4.5)
LSM <8 kPa	581 (83%)
LSM 8–12 kPa	97 (14%)
LSM >12 kPa	21 (3%)

ALP: alkaline phosphatase; BMI: body mass index; DM: diabetes mellitus; GGT: serum gamma-glutamyltransferase; HDL: high-density lipoprotein; DM: diabetes mellitus; IQR: interquartile range. * Number of missing values: 10 for GGT, 43 for bilirubin, 60 for ALP, 59 for albumin, 58 for cholesterol, 136 for HDL, 63 for triglycerides.

**Table 3 ijms-23-12489-t003:** Odds ratios (OR), and corresponding 95% confidence intervals (CI), of increased liver stiffness (LSM ≥ 8 kPa) by transient elastography according to selected demographic, laboratory, metabolic and clinical factors in 699 NAFLD patients with FIB-4 < 1.3.

	LSM ≥ 8 kPa, *n* (%) (*n* = 118)	Unadjusted OR (95% CI)	Multiple-Adjusted OR * (95% CI)
Age (five-years increase)		1.13 (1.40–1.23)	
Age, categories			
<60	70/517 (13.5)	Ref	Ref
≥60	48/182 (26.4)	2.29 (1.51–3.46)	1.96 (1.19–3.23)
Sex			
Female	44/301 (14.6)	Ref	Ref
Male	74/398 (18.6)	1.33 (0.89–2.01)	1.33 (0.85–2.08)
DM			
No	64/527 (12.1)	Ref	Ref
Yes	54/172 (31.4)	3.31 (2.19–5.01)	2.59 (1.63–4.13)
Hypertension			
No	57/440 (13.0)	Ref	Ref
Yes	61/259 (23.6)	2.07 (1.39–3.09)	1.06 (0.66–1.70)
Dyslipidemia			
No	64/418 (15.3)	Ref	
Yes	54/281 (19.2)	1.32 (0.88–1.96)	
BMI (kg/m^2^), one-point increase		1.12 (1.08–1.17)	
BMI (kg/m^2^), categories			
<27	27/281 (9.6)	Ref	Ref
≥27	91/418 (21.7)	2.62 (1.65–4.15)	2.17 (1.33–3.56)
GGT, 10 UI/L increase		1.05 (1.02–1.08)	
GGT (UI/L), categories			
<25	16/179 ^ (8.9)	Ref	Ref
≥25	97/510 (19.0)	2.39 (1.37–4.19)	2.68 (1.49–4.84)
Bilirubin, 1 mg/dL increase		0.73 (0.43–1.26)	
ALP, 10 UI/L increase		1.01 (0.97–1.04)	
Albumin, 1 g/dL increase		0.61 (0.34–1.10)	
Total cholesterol, 10 mg/dL increase		0.97 (0.92–1.02)	
HDL cholesterol, 10 mg/dL increase		0.90 (0.79–1.03)	
Triglycerides, 10 mg/dL increase		1.01 (0.98–1.03)	

ALP: alkaline phosphatase; BMI: body mass index; GGT serum-gamma-glutamyltransferase; HDL: high-density lipoprotein; DM: diabetes mellitus; Ref: reference category. * The model included all the factors simultaneously. ^ The sum does not add up to the total because of some missing values.

**Table 4 ijms-23-12489-t004:** Odds ratios (OR), and corresponding 95% confidence intervals (CI), of increased liver stiffness (LSM ≥ 8 kPa) by transient elastography according to the number of risk factors for LSM ≥ 8 kPa * in 699 NAFLD patients with FIB-4 < 1.3.

	LSM ≥8 kPa n ^ (%)	Unadjusted OR (95% CI)	Sex-adjusted OR (95% CI)
No. of risk factors *			
0	3/46 (6.5)	Ref	Ref
1	12/212 (5.6)		
2	44/276 (15.9)	3.07 (1.66–5.67)	3.05 (1.65–5.63)
3	36/116 (31.0)	7.29 (3.79–14.01)	7.29 (3.79–14.02)
4	18/39 (46.2)	13.89 (6.13–31.45)	14.46 (6.36–32.91)

CI: confidence interval; OR: odds ratio; Ref: reference category. * Risk factors were: age ≥ 60 years; body mass index ≥ 27 kg/m^2^; diabetes mellitus; serum-gamma-glutamyltransferase ≥ 25 UI/L. ^ The sum does not add up to the total because of some missing values.

## Data Availability

The data presented in this study are available on request from the corresponding author. The data are not publicly available because they contain personal information of the patients.

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
