# Peer review of "Accuracy of FIB-4 to Detect Elevated Liver Stiffness Measurements in Patients with Non-Alcoholic Fatty Liver Disease: A Cross-Sectional Study in Referral Centers"

_ijms, 2022, doi:10.3390/ijms232012489_

Round 1
Reviewer 1 Report
Dear Editor,
I have read with great interest the manuscript submitted by Viganò et al., entitled: "Accuracy Of FIB-4 To Detect Elevated Liver Stiffness Measurements In Patients With Non Alcoholic Fatty Liver Disease: A Cross Sectional Study In Referral Centers". There are some important issues that need to be addressed before further processing:
Introduction
The authors should clarify some lines. FIB-4 was developed for viral infections. In NASH, serum ALT levels can be altered and therefore affect (and overestimate FIB-4) score. Also, the authors should report that they are comparing and non-invasive indirect fibrosis testing and a "direct" fibrosis testing. I suggest citing the interesting results from the Bagnacavallo Study - DOI: https://doi.org/10.1016/j.aohep.2020.04.003).
Results
Check the M&M section.
Discussion and conclusions
The discussion is well organized, however the authors should highlight possible confounding factors that can affect liver stiffness measurement in patients with NAFLD (e.g. skin-to-liver distance, DOI: 10.3390/diagnostics10100795).
Materials and Methods
What test was used to study the distribution of data? The way logistic regression was performed appears to be not correct. First of all, it is not correct to report that the analysis is adjusted for a variable by just removing the aformentioned variable. Also, when performing the analysis univariately it is correct to study each variable singularly. When performing a multivariate model, you should also report possible model calibration (eg. AIC, BIC, etc.) and not only discrimination ability (i.e. AUROC). Finally is not clear in the text the differences of the training and test set. Also, was the p-value two tailed?
Author Response
Introduction
The authors should clarify some lines. FIB-4 was developed for viral infections. In NASH, serum ALT levels can be altered and therefore affect (and overestimate FIB-4) score. Also, the authors should report that they are comparing and non-invasive indirect fibrosis testing and a "direct" fibrosis testing. I suggest citing the interesting results from the Bagnacavallo Study - DOI: https://doi.org/10.1016/j.aohep.2020.04.003).
We agree with the Reviewer that FIB-4 was initially developed for viral infections and that serum ALT levels can be altered in patients with NASH, affecting and overestimating the FIB-4 results. We added a sentence at page 2 line 50 and at page 6 lines 175-176 reporting that FIB-4 was initially developed in HIV/HCV co-infected patients. However, to date FIB-4 has been validated in different etiologies of liver disease, although most of which may lead to an increase in transaminases levels affecting the FIB-4 results. In our study AST and ALT mean were 29 and 36 UI/L, respectively with only 38% of patients having ALT >ULN. With all these caveats, the recent EASL CPG on NITs, in patients with metabolic co-factors at risk of liver disease, FIB-4 was chosen to identify patients requiring referral to the specialist Liver Clinics and our study tried to explore the accuracy of this strategy in NAFLD patients. According to reviewer suggestion we added a sentence in the discussion about the comparison between a non-invasive indirect fibrosis testing and a "direct" fibrosis testing also including the paper by Foschi et al. (ref 30) reporting the lack of association between LSM and FIB-4 (page 7, lines 215-2).
Results
Check the M&M section.
According to reviewer suggestion we check the M&M section.
Discussion and conclusions
The discussion is well organized, however the authors should highlight possible confounding factors that can affect liver stiffness measurement in patients with NAFLD (e.g. skin-to-liver distance, DOI: 10.3390/diagnostics10100795).
We agree with reviewer that LSM may have many confounders. The most important is related to overweight or obesity. However, in our cohort the 36 patients (2%) who had an unreliable Fibroscan result were excluded (see the added flow chart in Figure 2). Giuffré et al. evaluated the LSM by point-shear wave elastography (ElastPQ protocol) with two different ultrasound machines, but this is a different method from the Transient Elastography by Fibroscan. According to the manufacturer, with Fibroscan the M probe should be used when the skin–liver capsule distance (SCD) is <25 mm, and the XL probe should be used when SCD is ≥25 mm. Having both probes (M and XL) we had the opportunity to use the best available probe for each individual patient thus reducing the bias related to the skin-to-liver distance in patients with NAFLD.
Materials and Methods
What test was used to study the distribution of data? The way logistic regression was performed appears to be not correct. First of all, it is not correct to report that the analysis is adjusted for a variable by just removing the aformentioned variable. Also, when performing the analysis univariately it is correct to study each variable singularly. When performing a multivariate model, you should also report possible model calibration (eg. AIC, BIC, etc.) and not only discrimination ability (i.e. AUROC). Finally is not clear in the text the differences of the training and test set. Also, was the p-value two tailed?
Continuous variables were checked for normal distribution by visual inspection of histograms and Kolmogorov–Smirnov tests. Normal and non-normal distributed continuous variables were presented, respectively, as mean ± SD, and median plus interquartile range; categorical data were presented as absolute frequencies and percentages. This was added to the statistical method section (page 9 lines 310-321). Data in Table 1 were also modified accordingly.
As for your second point, the primary objective of the analysis was to assess the association of selected variables with increased LSM among patients with NALFD and FIB-4 <1.3. Firstly, we considered each variable singularly, using a logistic model including as independent variable only the exposure under consideration. Several unadjusted models were therefore fitted, one for each factor. Results (i.e., odds ratios with 95% confidence intervals) from such analysis are presented in the “Unadjusted OR (95% CI)” column of Table 3. Secondly, we ran a multivariable logistic regression model, including simultaneously as independent variables those significantly associated with the outcome (i.e., LSM >8 kPa) in the unadjusted analyses, plus sex. Thus, the ORs presented in the last column of Table 3 derive from a unique model including age, gender, DM, hypertension, BMI and GGT as independent factors. This was better explained in the statistical analysis section.
As secondary complementary analysis (as specified in the manuscript test), we attempted to developed a model for individualized prediction of the outcome (i.e., LSM>8 kPa) based on standard methods for the selection of the predictors, evaluation of the model performance and (internal) validation. We used as measure of model calibration the calibration plot, comparing the average model prediction with the observed proportion of increased LSM across deciles of risk (see Figure 1, Panel B) [the following reference was added: Steyerberg and Vergouwe. Eur Heart J. 2014;35(29):1925-31]. We now added, as suggest by the Reviewer, the model AIC in the Results (page 6, lines 148-149).
As for your point on “training” and “test” sets”, we did not used a split sample approach, which is considered sub-optimal by many authors (see for example the reference previously quoted). We used the full database for model development and the bootstrap resampling (200 bootstrap samples, of the same size as the original dataset, were generated by drawing with replacement) for model internal validation. The bootstrapping method was reported to be the best method for producing efficient estimates that have low bias and low variability (ref: Steyerberg et al. Internal validation of predictive models: efficiency of some procedures for logistic regression analysis J. Clin. Epidemiol., 54 (2001), pp. 774-781). This was explained in the statistical analysis section.
Finally, P-values were two tailed; we had reported that “All tests were two-sided”; we reworded the sentence as follows, to make it more clear: “P-values were two-tailed, with a p<0.05 considered statistically significant.”
Reviewer 2 Report
Authors aimed to assess concordance between FIB-4 and Liver Stiffness Measurement (LSM) in patients referred to 2 liver centers for ultrasound-based diagnosis of NAFLD. The topic is interesting, however, as LSM is not the gold standard for diagnosis of liver fibrosis, it seems difficult to draw these conclusions. And there are several concerns as follows
1. Please present the main results in Table 2: 3% of patients with FIB-4 <1.3 showed an LSM indicative of severe fibrosis (>12 kPa), while 14% of them showed an LSM compatible with intermediate fibrosis (8-12 kPa).
2. Among a total of 1338 patients with NAFLD, 32% showed LSM≥8 kPa, with presumed liver fibrosis. I think this prevalence higher than those other population. Please discuss this issue.
3. What do you think is the cause of these discrepancies in FIB4 and LSM? Please discuss especially, regarding false positive FIB4.
4. Method: Please add a flow chart with showing numbers of excluded patients.
5. Authors showed the predictive model formulation and the AUC-ROC of the model was 0.743, indicating a reasonable accuracy of the model in discriminating between patients with and without increased LSM. However, external validation is essential to check whether this prediction formula works well.
6. Page 7, line 191-192: there may be an error.
Author Response
1 Please present the main results in Table 2: 3% of patients with FIB-4 <1.3 showed an LSM indicative of severe fibrosis (>12 kPa), while 14% of them showed an LSM compatible with intermediate fibrosis (8-12 kPa).
According to reviewer suggestion we add all these informations on LSM in patients with FIB-4 <1.3 in Table 2
2. Among a total of 1338 patients with NAFLD, 32% showed LSM≥8 kPa, with presumed liver fibrosis. I think this prevalence higher than those other population. Please discuss this issue.
Our study includes a high rate of patients with metabolic comorbidities such as arterial hypertension (49%), dyslipidemia (45%), diabetes (42%) and overweight (80%) and these conditions can influence the rate of patients with more severe disease. Also in the Graupera study the prevalence of LSM ≥8 kPa, as indirect evidence of significant liver fibrosis, among 1150 patients with metabolic liver disease was 29%.
3. What do you think is the cause of these discrepancies in FIB4 and LSM? Please discuss especially, regarding false positive FIB4.
We agree with the Reviewer about the limited accuracy of FIB-4 in this cohort of patients and more generally in patients with metabolic liver disease. The test was not designed for patients with metabolic liver disease and that there are likely confounders affecting both false negative and positive results. Although we identified an increasing prevalence of false negative FIB-4 among older patients with DM or high BMI, we did not identify predictors of the false positive FIB-4 results.
4 Method: Please add a flow chart with showing numbers of excluded patients
We add a flow chart with patient’s disposition (Figure 2)
5. Authors showed the predictive model formulation and the AUC-ROC of the model was 0.743, indicating a reasonable accuracy of the model in discriminating between patients with and without increased LSM. However, external validation is essential to check whether this prediction formula works well.
We agree with the Reviewer on the importance of external validation towards clinical implementation of prediction models, and, in line with this, we did not emphasize findings from the prediction model nor encourage clinicians to adopt our model for individualized prediction of increased LSM in patients with NAFLD and low FIB-4. We included now the lack of external validation among study limitation (page 8, lines 262-2)
6. Page 7, line 191-192: there may be an error.
We amended the typo (see page 7, lines 191-197)
Reviewer 3 Report
The manuscript is well conceived, but it proves the opposite of the hypothesis. Although this study was well written, there were serious objections in the study as written below. It is unlikely that FIB-4 can replace LSM. High false negative results with FIB-4<1.3 and LSM≥8 are relatively high. AUC-ROC is relatively good at FIB-4 <1.3 and LSM≥8. In the text the materials and methods should come before the discussion. I believe that FIB-4<1.3 can not be the only screening method for NAFLD. From this manuscript it is not possible to distinguish what is the exact purpose of your survey (FIB-4 < 1.3; this is a clear evidence of the NAFLD diagnosis without performing LSM). In addition, FIB-4 and LSM should always be performed together in the diagnosis of NAFLD.
Author Response
The manuscript is well conceived, but it proves the opposite of the hypothesis. Although this study was well written, there were serious objections in the study as written below. It is unlikely that FIB-4 can replace LSM. High false negative results with FIB-4<1.3 and LSM≥8 are relatively high. AUC-ROC is relatively good at FIB-4 <1.3 and LSM≥8. In the text the materials and methods should come before the discussion. I believe that FIB-4<1.3 can not be the only screening method for NAFLD. From this manuscript it is not possible to distinguish what is the exact purpose of your survey (FIB-4 < 1.3; this is a clear evidence of the NAFLD diagnosis without performing LSM). In addition, FIB-4 and LSM should always be performed together in the diagnosis of NAFLD.
The aim of the study was to evaluate the accuracy of FIB-4 to detect elevated LSM in patients with NAFLD. The European Association for Study of the Liver (EASL) strongly recomemended the use in primary care of FIB-4 in patients with metabolic risk factors due to the high risk of liver fibrosis in order to improve risk stratification and linkage to care. In our study, among the 699 patients with FIB-4 <1.3: 581 (83%) patients showed LSM values below the 8 kPa, but 118 (17%) patients had LSM suggestive of intermediate-advanced fibrosis (14% of them showed an LSM compatible with intermediate fibrosis (8-12 kPa) and 3% showed a LSM indicative of severe fibrosis (>12 kPa).
We agree with the reviewer that, due to the high false negative FIB-4 results, it is unlikely that FIB-4 can replace LSM and most important that FIB-4 cannot be the only screening method for NAFLD and maybe FIB-4 and LSM should always be performed together in patients with NAFLD.
In an attempt to overcome the limits of FIB-4 in such patients we developed a predictive model. Each unit increase in age, BMI and GGT increases the natural logarithm of the odds of LSM ≥8 kPa respectively by 0.022, 0.127 and 0.007; this was increased by 1.03 in diabetics. Model AIC was 548.2, and AUC-ROC was 0.743, indicating a reasonable accuracy of the model in discriminating between patients with and without increased LSM (Figure 1, panel A).
Round 2
Reviewer 1 Report
Dear Editor,
I have read with great interest the revised version of the manuscript entitled "Accuracy Of FIB-4 To Detect Elevated Liver Stiffness Measurements In Patients With Non Alcoholic Fatty Liver Disease: A Cross Sectional Study In Referral Centers" by Viganò et al.
The authors have replied to the reviewer's comments. I believe the manuscript can now be accepted for publication in IJMS.
Author Response
We thank the reviewer for his comment.
Reviewer 2 Report
Authors mentioned the presence of comorbidities as DM, arterial hypertension and dyslipidemia in the Method section. However, information on arterial hypertension is not availabe in the Table 1 and Table 2. Please add.
Author Response
We thank the reviewer for the comment. Information about arterial hypertension are available in table 1 and table 2.